# On Developing a Machine Learning-Based Approach for the Automatic Characterization of Behavioral Phenotypes for Dairy Cows Relevant to Thermotolerance

Oluwatosin Inadagbo [1], Genevieve Makowski [2], Ahmed Abdelmoamen Ahmed [1,*] and Courtney Daigle [2]

[1] Department of Computer Science, Prairie View A&M University, Prairie View, TX 77446, USA; oinadagbo@pvamu.edu
[2] Department of Animal Science, Texas A&M University, College Station, TX 77843, USA; gmakowski@tamu.edu (G.M.); courtney.daigle@ag.tamu.edu (C.D.)
* Correspondence: amahmed@pvamu.edu

**Abstract:** The United States is predicted to experience an annual decline in milk production due to heat stress of 1.4 and 1.9 kg/day by the 2050s and 2080s, with economic losses of USD 1.7 billion and USD 2.2 billion, respectively, despite current cooling efforts implemented by the dairy industry. The ability of cattle to withstand heat (i.e., thermotolerance) can be influenced by physiological and behavioral factors, even though the factors contributing to thermoregulation are heritable, and cows vary in their behavioral repertoire. The current methods to gauge cow behaviors are lacking in precision and scalability. This paper presents an approach leveraging various machine learning (ML) (e.g., CNN and YOLOv8) and computer vision (e.g., Video Processing and Annotation) techniques aimed at quantifying key behavioral indicators, specifically drinking frequency and brush use-behaviors. These behaviors, while challenging to quantify using traditional methods, offer profound insights into the autonomic nervous system function and an individual cow's coping mechanisms under heat stress. The developed approach provides an opportunity to quantify these difficult-to-measure drinking and brush use behaviors of dairy cows milked in a robotic milking system. This approach will open up a better opportunity for ranchers to make informed decisions that could mitigate the adverse effects of heat stress. It will also expedite data collection regarding dairy cow behavioral phenotypes. Finally, the developed system is evaluated using different performance metrics, including classification accuracy. It is found that the YoloV8 and CNN models achieved a classification accuracy of 93% and 96% for object detection and classification, respectively.

**Keywords:** dairy cows; behavioral phenotypes; thermotolerance; heat stress (HS); artificial intelligence; machine learning (ML); computer vision

## 1. Introduction

The dairy industry is increasingly facing grand challenges due to climatic changes [1,2], with heat stress being one of the most significant environmental factors affecting dairy cattle [3]. Projected climatic trends indicate a troubling forecast for dairy production in the United States, with anticipated decreases in milk production due to heat stress expected to reach significantly low levels by 2080 [4]. Despite current cooling efforts, these losses are juxtaposed against the increasing need to identify judicious uses of natural resources, including water [5].

The adverse effects of heat stress on cattle include diminished milk production, decreased reproductive capabilities, heightened susceptibility to diseases, and potentially increased mortality rates [6]. These consequences affect productivity and translate into considerable annual economic losses, estimated at billions of dollars. The ability of dairy cows to withstand heat, termed thermotolerance, is affected by a combination of physiological and behavioral aspects. These traits are significantly heritable and exhibit considerable

variation across individual cows, complicating the challenge of effectively managing heat stress within dairy herds [4].

The complexity of interpreting cow behavior in relation to heat stress is heightened by genetic diversity influencing their capacity to manage thermal stress [7]. In particular, drinking and environmental enrichment use are challenging behaviors to quantify but may be the most informative behaviors for characterizing how individuals cope with heat stress. The intricate nature of thermal adaptation necessitates integrating sophisticated, non-invasive measures into the genetic selection process to enhance the thermotolerance of dairy herds. Thus, interpreting cow behavior in response to heat stress is complicated because some cows are genetically better suited to cope with heat stress. In contrast, others are behaviorally flexible in dealing with thermal challenges [8].

Integrating automated, non-invasive phenotypic indicators of thermotolerance into genetic selection decisions using metrics relevant to thermotolerance is necessary [4]. However, the existing monitoring methods of heat stress are labor-intensive and often fail to provide timely data [9]. Furthermore, the increase in dairy size is juxtaposed against the need to monitor individual animals using fewer employees. Because of this paradigm, there is a persistent need to develop new strategies and technologies for monitoring individual animals in large groups [1].

Cattle have a variety of inherent traits that can be used to identify unique individuals, including coat patterns, iris patterns, retinal patterns, facial features, and muzzle patterns [8]. Holstein cattle, a common breed of dairy cattle in the US, are easily recognizable by their distinctive black-and-white patterns. Each cow's pattern of spots is unique, making this morphological feature a useful biometric tool for individual identification [10].

This paper presents an approach to address these challenges by developing a system that provides real-time and automated monitoring of dairy cows milked in a robotic milking system using artificial intelligence (AI) and computer vision technologies [11]. First, we present an imagery data collection and processing approach that automatically detects and quantifies the drinking and brush use behavior of Holstein cattle dairy cows using their coat patterns. Second, the presented approach performs the fundamental research needed to enable the characterization and development of non-invasive behavioral phenotypes indicative of a cow's ability to withstand heat stress. These behaviors (i.e., brush use and drinking behavior) are integral to maintaining homeostasis, particularly during heat stress [4]. Monitoring an animal's use of these resources provides insight into an animal's inherent water efficiency (e.g., drinking behavior), temperament (e.g., resource use frequency, circadian pattern, and plasticity to environmental conditions), and motivation to engage in pleasurable behaviors (e.g., brush use) that ultimately promotes animal welfare [3].

To validate the applicability of the proposed approach, we captured a video dataset consisting of 3421 videos with a total duration of 24 h of continuous recording of dairy cows housed at the T&K Dairy, a commercial dairy partner in Snyder, Texas. Figure 1 shows examples from the collected video dataset. As shown in the video snapshots, the cows are housed in a single free-stall barn that is divided into six pens (*n* = 180 cows/pen). Each pen provides cattle with access to four water troughs evenly placed throughout the barn. Near three of the water troughs within the pen, cattle have access to an automatic rotating cattle brush that is mounted to the barn.

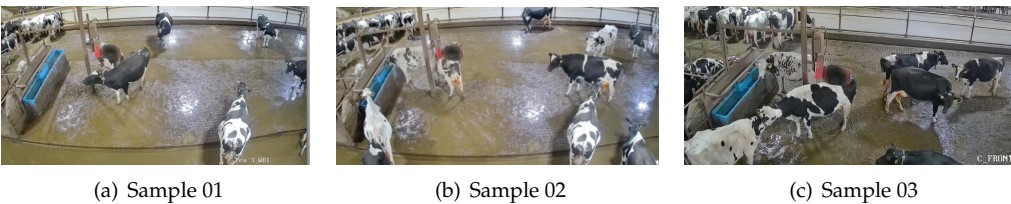

(a) Sample 01        (b) Sample 02        (c) Sample 03

**Figure 1.** Sample examples from our video dataset.

The individual cows that appear in this video dataset were identified using clustering algorithms (e.g., K-means [12]) to assign unique identifiers to individual cows, converting raw visual video streams into structured and analyzable formats stored in a relational database. Then, we utilized ML-based object detection models (e.g., YoloV8 [13]) to accurately recognize individual cows using their coat patterns within the complex farm environments. A Convolutional Neural Network (CNN) model [14] is trained using the extracted cow objects to classify each cow to a particular cluster in our database, which is used in conjunction with the DeepSORT algorithm [15] to track cow activities and provides accurate quantification of watering and brush use behaviors. Finally, a user-friendly GUI interface is developed to enable system users to utilize the developed system conveniently.

This paper makes the following contributions. First, we present a machine learning approach that can automatically capture, process, and visualize massive video datasets to characterize behavioral phenotypes for dairy cows relevant to thermotolerance. Second, a novel object-tracking module is proposed to detect moving cows' behavior in real-time CCTV footage videos. Third, this paper presents a GUI interface on top of a pipeline of ML models and computer vision algorithms (i.e., K-means, YoloV8, CNN, and DeepSORT) to allow ranchers to interact with the developed system conveniently using a web-based GUI.

## 2. Related Work

Machine learning (ML) coupled with computer vision [14,16–19] has already enabled game-changing capabilities of robotic milking systems by providing the ability to enhance dairy cow health management by automating the detection and analysis of heat stress behaviors using CCTV footage videos [7]. ML and computer vision have been used in the literature for a wide variety of functionalities in the dairy cattle domain, including the identification of individual animals [18], analysis of cow behaviors such as feeding [20] and standing and lying [7], and the detection of health indicators such as lameness [21] and body condition score [17].

Fuentes et al. [11] studied the use of ML and computer vision to identify the age of cattle based on their facial features. The face location was detected in still frames isolated from recorded video using YOLOv5. The authors used the MobilenetV2 tool to extract the face's vector of 128 features and aligned it with Resnet18. The extracted feature vector is then fed into an ML model to predict the animal's age accurately. Despite the similarity in the scope, our approach uses different methodologies in utilizing a pipeline of ML and clustering algorithms to identify individual cows based on side-angle images of their coat patterns.

In [7], the authors used computer vision techniques to detect the lying behavior of dairy cows in a freestall barn. Similar to our work, the authors used a combination of YOLOv5x and DeepSORT to identify and track cows using individual bounding boxes for each cow. Changes in the properties of the bounding boxes were used to identify the start and end of positional change events (i.e., lying down and standing from a lying position). However, no attempt has been made to identify cattle based on their biometrics. While the bounding boxes are used to detect behaviors, the behaviors are detected using the changing properties of a single box, unlike our presented approach, which involves two bounding boxes overlapping.

Gupta et al. [22] used the YOLOv4 model to identify cattle by breed. The YOLOv4 model was trained using a custom dataset of eight cattle breeds. The authors evaluated the model using an intersection over union metric, precision–recall curves, a confusion matrix, an overall accuracy equation, and Cohen's kappa. The model was experimentally proven more effective with smaller and high-resolution images. When comparing the YOLOv4 model to other models used for breed detection (e.g., faster RCNN, SSD, and YOLOv3), it is found that YOLOv4 improved the performance of the three models.

Another work presented in [18] attempted to develop a cattle identification method based on coat patterns. Videos were captured from a top angle, resulting in a top-down image of the cow's back. A Mask R-CNN model was used to identify the patterned region

of the cow and extract pattern features from the frames of the video, after which a Support Vector Machine (SVM) was used to identify cows based on their pattern features. The resulting system had an accuracy of 98.67%. While this project and ours focus on identifying cattle by coat pattern, the methodologies used vary significantly. The previous work uses a top-down view of the cow in contrast to ours, which uses a side view.

Wang et al. [14] used a 3D-based CNN (E3D) algorithm to classify five cow behaviors in a clip of video: standing, walking, drinking, feeding, and lying down. Videos captured from cattle pens were split into short segments, each containing one of the behaviors of interest. The E3D algorithm comprised several modular parts: a 3D convolution module, a SandGlass-3D module, and an ECA module. The 3D convolution module extracted features from the still video frames, which were then put through the SandGlass 3D module to identify spatial and temporal properties. Background information from the videos was screened and removed by the ECA module. A 3D pooling layer and the Softmax function were used as the final processing steps to compress the behavioral features and perform the behavioral classification, respectively. The proposed model achieved high accuracy in detecting and classifying cow behaviors. This project adopted a different approach from ours, though both were based on a CNN model. Both projects also identify multiple behaviors with the same algorithm. However, this work focuses solely on behavior detection and does not attempt to identify cattle as individuals.

Another study presented in [23] achieved acceptable results in detecting the behaviors of dairy cows. The authors focused on developing a deep learning model called Res-DenseYOLO, which is an improvement of the YOLOv5 model by incorporating DenseNet and residual network structures to enhance feature extraction, for the automatic recognition of dairy cow behaviors, specifically standing, lying, eating, and drinking. However, this work has not implemented the unique identification of individual cows or continuously tracked the duration of behaviors.

Despite the previous success of cattle identification using computer vision to identify coat patterns, the existing work has notable limitations [17,24]. The used imagery dataset has a small field of view, often set up where cattle walk through narrow passages with limited ability to turn [8,10,25]. Additionally, the lighting is constant, and there may be only one or a few cattle in the frame at a time, all of which simplify the task of identifying cattle by computer vision but limit the potential applications in a busy barn. In contrast, our approach is designed to identify cattle at a distance and in an open space within a broad view frame.

In summary, the existing work focusing on the automatic characterization of behavioral phenotypes for dairy cows used different approaches for cattle identification [11,17,18] and behavioral monitoring [7,11,22] via computer vision and ML. However, none of them have successfully combined these two objectives into a single platform and conveniently provided a user-friendly GUI interface to the system. To the best of our knowledge, our approach represents the first step to building a system that automatically identifies dairy cattle based on biometric features and monitors their behavior of interest based on interactions with other objects in the barn (i.e., water troughs and brush stations).

## 3. Design

### 3.1. Dataset Collection

Figure 2 illustrates the camera placement in the barn at a low angle to capture the side of the cows. As shown in the schematic figure, each barn at the T&K Dairy is fitted with Safevant, Safesky, and 1080P Isotect wireless security cameras that continuously capture individual cow behavior at the waterers and the brushes throughout the 45-day observation period. Cows are milked using a Lely Robotic Milking System, equipped with 18 robots and 3 robots per pen, that milks the cows twice daily.

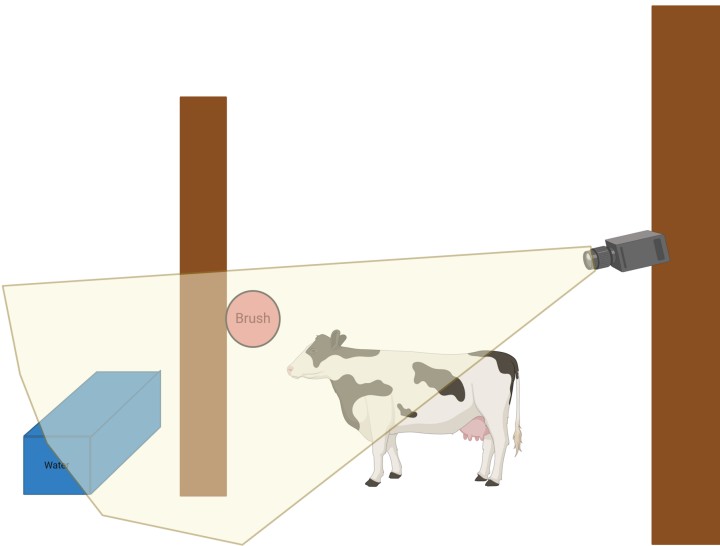

**Figure 2.** Camera placement in the barn.

Cows are cooled using multiple different strategies. The barn is equipped with fans, sprinklers, and foggers. The sprinklers begin running, at rates of one-minute durations, in a round-robin system across all pens when the air temperature in the barn exceeds 74 °F. Thus, each pen will have the sprinklers turned on for one minute at least ten times per hour until the temperature falls below 74 °F. When the temperature in the barn exceeds 80 °F, the fogger system will begin operating and will continue until the temperature drops below 80 °F.

Several variables are collected using the robotic system and recorded in the Lely management software Time for Cows (T4C). This project is of specific interest in milk production, yield, maximum milk speed, dead milking time, and robot behavior (i.e., visit, rejection, and fetch frequency). A subset of focal cows ($n = 96$; 16 cows/pen) that are 45–90 DIM were monitored for a 45 d period.

We captured a video dataset on 12–13 March 2023, consisting of 3421 videos with a total duration of 24 h of continuous recording of the cows. These videos were recorded in DAT format. We converted them to MP4 format using the FFmpeg conversion tool [26]. This preprocessing step was necessary because the MP4 format has high compression and compatibility with numerous multimedia applications, making it the preferred choice for ensuring seamless playback and processing.

These video recordings were used quantify drinking behavior and brush use behavior. When individual identification is required on the video recordings, each dairy cow has a unique coat color spotting pattern that can be used for individual identification. During the time that individuals were fitted with pedometers, their drinking and brush use behavior (frequency, duration, circadian pattern, displacements) were decoded from video recordings. While this is possible using manual decoding methods, the development of automatic ML-based methods can expedite data collection, knowledge creation, and results implementation.

### 3.2. Methodology

The practical side of the proposed approach is to build machine vision and ML methods to support the automatic acquisition and processing of imagery data needed to develop behavioral phenotypes for dairy cows relevant to thermotolerance. The foundational work aims to understand the principles underlying such systems and inform the design and implementation decisions about them.

Figure 3 shows the system architecture of the proposed approach, which is divided into four layers. Layer 1 shows the video preprocessing phase, which involves slicing the collected video dataset (i.e., 3421 videos) into individual frame images using a Python 3.12.4 script leveraging the FFmpeg framework [26] at predefined intervals. We then extracted

1961 cow objects to train the cow clustering algorithm and CNN model. The Roboflow tool [27] was utilized to annotate the cow, brushing tool, and waterer objects to train the detection and segmentation models.

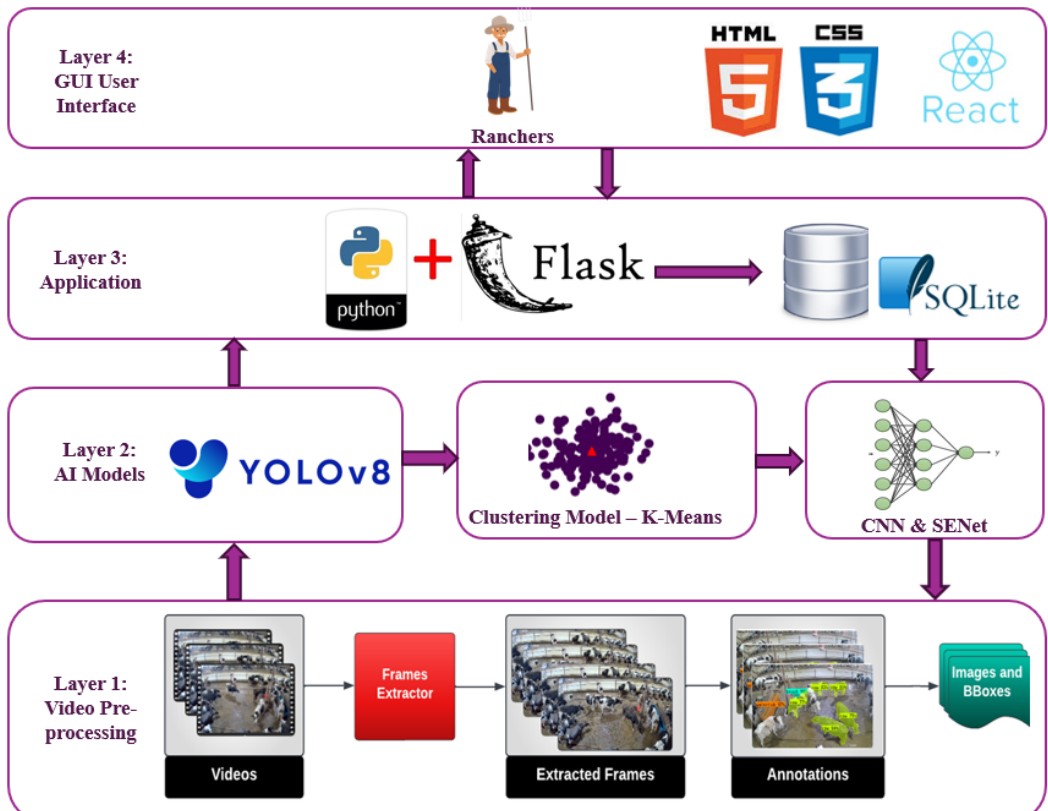

**Figure 3.** System architecture.

Layer 2 shows the cow detection and segmentation module using the YOLOv8 model, the cow clustering module using the K-means model, and the cow identification module using the CNN and SENet models [28]. Layer 3 describes the application layer implemented using the Python Flask Framework [29] and SQLite version 3.46 database engine [30] to build a GUI web-based app that allows system users, shown in layer 4, to use the system conveniently.

Using unsupervised and supervised ML models alongside algorithmic tracking and behavior analysis [24], we utilized a pipeline approach for processing the video dataset where data flows from one layer to the next. Figure 4 shows the different phases of cow detection, clustering, identification, and tracking behaviors of interest.

### 3.2.1. Cow Detection and Segmentation Using YoloV8

The YoloV8 model is trained using a custom imagery dataset to accurately detect and segment the cow objects in the video frames. The YoloV8 model is a deep learning algorithm used for its high-performance detection of real-time objects within video streams. Upon receiving video input, YoloV8 processes the frames to identify and locate the cow, water tank, and brushing tool objects, assigning bounding boxes around them.

After detecting the objects of interest (i.e., cow, water tank, and brushing tool objects), we used a cropping tool to extract the bounding boxes generated by YoloV8 containing these objects from the frames. This extraction process is vital to isolate objects of interest from their background, allowing for cleaner data input into the next clustering phase. This process is summarized in Algorithm 1.

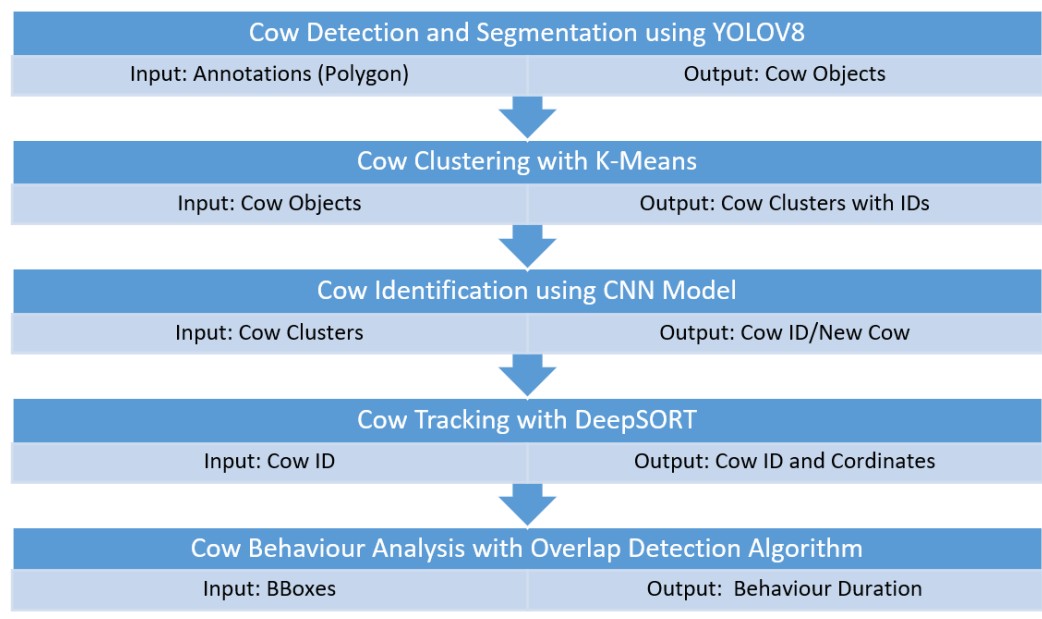

**Figure 4.** Phases of cow detection, clustering, identification and tracking behavior.

---

**Algorithm 1** Cow detection and segmentation using YoloV8.

---

1: Import necessary libraries and define classes
  SAMMaskGenerator
2: Initialize model with type, checkpoint, and device
3: **function** GENERATE_AND_SAVE_MASK(*image_path, save_all, save_rgba*)
4:    Load and process the image to RGB
5:    Generate masks and sort by area
6:    Save masks and optionally save RGBA images
7: **end function**
8: **function** PROCESS_IMAGES(*images_path, mask_generator, yolo_model, output_dir*)
9:    **for** each image in directory **do**
10:      Generate and save mask for each image
11:      Detect objects using the YOLO model
12:      Save images with detected objects
13:    **end for**
14: **end function**
15: **procedure** MAIN(*image_folder, sam_checkpoint, yolo_model_path, output_dir*)
16:    Initialize mask generator and YOLO model
17:    Process images in the specified folder
18: **end procedure**

---

We used the `TaskAlignedAssigner` class to improve the model's performance by effectively matching the predicted bounding boxes with ground truth boxes. In particular, it calculates a score *s* for each predicted box, as follows.

$$s = \gamma^m + \eta^n \tag{1}$$

where $\gamma$ is the prediction score corresponding to the ground truth category, $\eta$ is the IoU of the prediction bounding box and the ground truth bounding box, and *m* and *n* are hyperparameters that weight the importance of the classification score and the IoU score, respectively.

`TaskAlignedAssigner` ensures that only these prediction scores, which are confident in their class predictions and accurate in their localization, are selected as positive samples.

This dual consideration helps the model learn more effectively from classification and localization tasks, leading to improved overall performance in object detection.

### 3.2.2. Cow Clustering Using K-Means

The extracted cow objects are then fed into a K-means clustering algorithm, an unsupervised learning algorithm that groups the cow objects into clusters based on their visual similarities. The K-means algorithm iteratively assigns each cow object to one of $K$ predefined clusters based on feature similarities, minimizing variance within the clusters and maximizing variance between them.

Algorithm 2 shows the steps of the cow clustering phase, which is divided into the following processes: (i) texture feature extraction using the Local Binary Pattern (LBP) method and (ii) creating color histograms by capturing and analyzing the color distribution in the cow images. We also used the Principal Component Analysis (PCA) method to reduce the feature dimensionality and focus on the most important features of input images.

As shown in the algorithm, we select an initial cluster of centroids, $K$, randomly selected from the data points. We then assign each data point to the nearest cluster centroid. For a given data point $x_i$ and centroid $\mu_j$, the assignment process is performed as follows:

$$c_i = \arg\min_j \|x_i - \mu_j\|^2 \tag{2}$$

where $c_i$ is the cluster assignment for data point $x_i$, and $\|x_i - \mu_j\|^2$ is the squared Euclidean distance between $x_i$ and $\mu_j$.

The K-means algorithm updates the centroids after each iteration by calculating the mean of data points assigned to each cluster, as follows:

$$\mu_j = \frac{1}{|C_j|} \sum_{x_i \in C_j} x_i \tag{3}$$

where $\mu_j$ is the new centroid of cluster $j$, and $C_j$ is the set of data points assigned to cluster $j$.

K-means tries to minimize the Within-Cluster Sum of Squares (WCSS) inertia objective function, which is defined as:

$$\sum_{j=1}^{K} \sum_{x_i \in C_j} \|x_i - \mu_j\|^2 \tag{4}$$

The algorithm keeps iterating between the assignment and updates steps until convergence, typically when the cluster assignments no longer change or the change in the objective function is below a certain threshold.

### 3.2.3. Cow Identification Using a CNN and SENet Model

We trained a Convolutional Neural Network (CNN) model enhanced with Squeeze-and-Excitation Network (SENet) layers [28] using the cow clusters generated from the clustering phase to detect the cow objects based on their features and behaviors of interest. The training process allows CNN to learn the nuanced differences between clusters by calculating a similarity score for each cow against the cluster centroids. If the score exceeds a predefined threshold, the cow is assigned the ID of that cluster; otherwise, the cow is flagged as potentially new or not belonging to any existing cluster. Algorithm 3 shows the steps of the cow identification phase using the CNN and SENet model.

---

**Algorithm 2** Cow clustering using K-means

---

1: **function** EXTRACT_FEATURES(*image_path*)
2:     Read image from the path
3:     **if** image is not read correctly **then**
4:         Print error and return an empty list
5:     **end if**
6:     Convert image to grayscale and apply histogram equalization
7:     Calculate LBP and generate a histogram
8:     Calculate color histograms for each channel
9:     Combine and normalize histograms **return** Combined histogram as a feature vector
10: **end function**
11: **function** ORGANIZE_IMAGES(*clusters, image_paths, output_dir*)
12:     **for** each cluster and image path **do**
13:         Create or verify the existence of a directory for cluster
14:         Copy the image to the corresponding cluster directory
15:     **end for**
16: **end function**
17: **function** CLEAR_DIRECTORY(*dir_path*)
18:     **if** directory exists **then**
19:         Remove directory and contents
20:     **end if**
21:     Create directory
22: **end function**
23: **function** VISUALIZE_PCA_VARIANCE(*pca*)
24:     Plot PCA explained variance
25: **end function**
26: **procedure** MAIN
27:     Define image directory and output directory
28:     Clear output directory
29:     **if** image directory does not exist **then**
30:         Print error and return
31:     **end if**
32:     List all image paths in the directory
33:     Extract features from each image
34:     Remove empty feature lists
35:     **if** no valid features **then**
36:         Print error and return
37:     **end if**
38:     Standardize features
39:     Apply PCA to reduce feature dimensionality
40:     Optional: Visualize PCA variance
41:     Cluster features using KMeans
42:     Organize images into clusters based on their assigned cluster label
43: **end procedure**

---

The `SENet` block enhances the feature extraction and representation of the trained cow images by dynamically recalibrating channel-wise feature responses.

First, `SENet` applies a convolution operation, $\delta$ to the input feature map $I$, as follows:

$$X = f_\delta(I) \tag{5}$$

where $f_\delta(I)$ represents the convolution operation and $X$ is the output feature map with dimensions $H \times W \times C$.

---

**Algorithm 3** Cow identification using CNN and SENet Models

---

1: Define image transformations for data augmentation
2: Load the training, validation, and testing datasets
3: **function** IMSHOW(*input*, *title*)
4:     Convert tensor to image
5:     Display image with title
6: **end function**
7: Define the architecture of the SENetBlock
8: Define the architecture of the CowIdentificationModel
9: Instantiate the model and transfer to the computing device
10: Define loss function and optimizer
11: Apply weight initialization to model
12: **function** EVALUATE(*model*, *loader*, *device*)
13:     Evaluate the model with data from the loader
14:     Calculate accuracy **return** accuracy
15: **end function**
16: **function** TRAIN_MODEL(*model*, *train_loader*, *val_loader*, *num_epochs*)
17:     **for** *epoch* ← 1 **to** *num_epochs* **do**
18:         Train model for one epoch
19:         Calculate validation accuracy
20:         **if** validation accuracy is improved **then**
21:             Save model state
22:         **end if**
23:     **end for**
24:     Print best validation accuracy
25: **end function**
26: Train the model
27: Load the best model and evaluate on the test set
28: **function** ACCURACIES_PER_CLUSTER(*loader*, *model*, *device*)
29:     Calculate accuracies per cluster **return** cluster accuracies
30: **end function**
31: Print accuracy per cluster for validation and test datasets
32: Visualize predictions of training images
33: **function** COMPUTE_CLUSTER_CENTROIDS(*dataset*, *model*, *device*)
34:     Compute centroids for each cluster in the dataset **return** centroids
35: **end function**
36: **function** PREDICT_IMAGE(*image_path*, *model*, *transform*, *centroids*, *threshold*)
37:     Predict the label of an image using the model and cluster centroids **return** predicted
    label and similarity
38: **end function**
39: Define additional transformations for the test images
40: Compute centroids for the training dataset
41: Predict label for a test image and check similarity
42: **function** CALCULATE_METRICS_MANUAL(*preds*, *labels*)
43:     Calculate precision, recall, and F1 score manually **return** precision, recall, F1
44: **end function**
45: Calculate and print precision, recall, F1 for validation and test sets

---

Then, the squeeze operation performs a global average pooling on *X* to generate a channel descriptor $\theta$, which is defined as:

$$\theta_c = \frac{1}{H \times W} \sum_{i=1}^{H} \sum_{j=1}^{W} X_{i,j,c} \tag{6}$$

where $\theta_c$ is the *c*th element of the descriptor $\theta \in \mathbb{R}^C$.

The excitation operation models the channel-wise dependencies using two fully connected layers with ReLU and sigmoid activations, as follows:

$$\mathbf{s} = \sigma(W_2 \cdot \text{ReLU}(W_1 \cdot \theta)) \tag{7}$$

where $W_1 \in \mathbb{R}^{\frac{C}{r} \times C}$ and $W_2 \in \mathbb{R}^{C \times \frac{C}{r}}$ are the weight matrices, $r$ is the reduction ratio, and $\sigma$ is the sigmoid function.

Finally, the recalibration step scales the original feature map $X$ by the channel-wise weights $s$, as follows:

$$Y_{i,j,c} = s_c \cdot X_{i,j,c} \tag{8}$$

where $Y_{i,j,c}$ is the recalibrated feature map.

### 3.2.4. Tracking Cow Behaviors of Interest Using DeepSORT

The DeepSORT algorithm is used to track the cow's behaviors of interest (i.e., drinking and brushing). The DeepSORT algorithm extends the SORT (Simple Online and Real-Time Tracking) algorithm by incorporating deep learning features for more accurate tracking in crowded and complex environments, which can track multiple objects in a video stream, handling challenges such as occlusion and reappearance.

DeepSORT uses the assigned cluster IDs generated from the previous CNN phase across the video frames and associates the recognized behaviors of interest to the individual cows throughout recorded videos. Algorithm 4 summarizes this process.

---

**Algorithm 4** Tracking cow behaviors of interest using DeepSORT

---

1: **Input:** video_path, output_video_path, model_path
2: **Output:** Annotated Video, Activity Videos, Cow Images
3: **procedure** INFERENCE(*video_path, output_video_path*)
4:     Initialize YOLO model with *model_path*
5:     Initialize DeepSort object tracker
6:     Create directories for cow activity videos and images
7:     Load cow identification model
8:     Define image transformations
9:     Open input video and prepare output video writer
10:     **while** video has frames **do**
11:         Read frame from video
12:         Detect objects using the YOLO model
13:         Update tracks with DeepSort tracker
14:         **for** each detected cow **do**
15:             Capture cow image
16:             Transform and classify cow image to predict cluster ID
17:             Save cow image and update the database with the cluster ID
18:             **if** cow is performing an activity **then**
19:                 Record activity duration
20:                 Generate and save activity video
21:                 Update database with activity information
22:             **end if**
23:         **end for**
24:         Write annotated frame to output video
25:     **end while**
26:     Close video writer and release resources
27: **end procedure**
28: **procedure** PREDICT_CLUSTER_ID(*image*)
29:     Transform image to tensor
30:     Predict cluster-ID using cow identification model
31:     **return** predicted cluster-ID and probability
32: **end procedure**

---

We used the Kalman filter to enhance the motion prediction. The Kalman filter predicts the current state $x_t$ of an object based on its previous state $x_{t-1}$, as follows:

$$x_t = F_{x_{t-1}} + B_{u_t} + w_t \tag{9}$$

where $F$ is the state transition matrix, $B$ is the control input matrix, $u_t$ is the control vector, and $w_t$ is the process noise.

The observation model updates the state with new measurements $z_t$, as follows:

$$z_t = H_{x_t} + v_t \tag{10}$$

where $H$ is the observation matrix and $v_t$ is the measurement noise.

The cost matrix $\psi$ combines the motion and appearance information to match detections to tracks for better data association:

$$\psi_{ij} = \lambda \cdot d(y_i, y_j) + (1 - \lambda) \cdot (1 - \mathrm{cosine}(f_i, f_j)) \tag{11}$$

where $d(y_i, y_j)$ is the Mahalanobis distance between the predicted state $y_i$ and the actual detection value $y_j$, $\mathrm{cosine}(f_i, f_j)$ is the cosine distance between the deep feature vectors $f_i$ and $f_j$, and $\lambda$ is a weight parameter to balance motion and appearance costs.

### 3.2.5. Cow Behavior Analysis with Overlap Detection Algorithm

The final phase in our pipeline system is quantifying the cow behaviors of interest using an overlap detection algorithm. In particular, we developed an algorithm that calculates the duration each cow spends drinking water or using the brush tool by measuring the overlap area between the bounding boxes of the cow and the water tank or brushing tool. This duration is then logged into the database, which monitors the cows' health indicators over time.

In Algorithm 5, the `Calculate the Coordinates of the Bounding Box` procedure describes the step of calculating the coordinates of the bounding box of an object of interest using the input parameters $x_{center}, y_{center}, width, height$, and converting the result to a top-left format (i.e., $left, top, right, bottom$).

Procedure `Check the Existence of Overlap between the Bounding Boxes` presents the functionality of checking the existence of overlap between the bounding boxes of a cow object with a water tank or brushing tool object. The two coordinates of the two boxes, $bbox1, bbox2$, are fed into the *calculate_overlap_area* function, which returns *True* if there is an overlap between $bbox1$ and $bbox2$; otherwise, it returns *False*.

In Procedure `Calculate the Overlap Area between Two Bounding Boxes`, we calculate the overlap area between the input bounding boxes ($bbox1, bbox2$) using their centroid (i.e., the center $(x, y)$ coordinates of the bounding box) as follows: $x\_overlap \times y\_overlap$. We then calculate the Euclidean distance between the centroids of bounding boxes as described in Procedure *Calculate the Euclidean Distance between the Centroids of Bounding Boxes*.

We then check the proximity of a target bounding box to other boxes in a video frame by comparing the Euclidean distance of the target bounding boxes with all identified boxes in the scene using a predefined threshold.

---

**Algorithm 5** Cow behavior analysis using the overlap detection algorithm

---

1: **procedure** CALCULATE THE COORDINATES OF THE BOUNDING BOX
2:     **Require** $x_{center}, y_{center}, width, height$
3:     $left \leftarrow \text{int}(x_{center} - width/2)$
4:     $top \leftarrow \text{int}(y_{center} - height/2)$
5:     $right \leftarrow left + \text{int}(width)$
6:     $bottom \leftarrow top + \text{int}(height)$
7:     **Return** $left, top, width, height, right, bottom$
8: **end procedure**

9: **procedure** CHECK THE EXISTENCE OF OVERLAP BETWEEN THE BOUNDING BOXES
10:     **Require** $bbox1, bbox2$
11:     $left1, top1, right1, bottom1 \leftarrow bbox1$
12:     $left2, top2, right2, bottom2 \leftarrow bbox2$
13:     **if** $left1 \geq right2$ OR $right1 \leq left2$ OR $top1 \geq bottom2$ OR $bottom1 \leq top2$ **then**
14:         **Return** $False, 0$
15:     **else**
16:         $overlap\_area \leftarrow \text{calculate\_overlap\_area}(bbox1, bbox2)$
17:         **Return** $True, overlap\_area$
18:     **end if**
19: **end procedure**

20: **procedure** CALCULATE THE OVERLAP AREA BETWEEN TWO BOUNDING BOXES
21:     **Require** $bbox1, bbox2$
22:     $left1, top1, right1, bottom1 \leftarrow bbox1$
23:     $left2, top2, right2, bottom2 \leftarrow bbox2$
24:     $x\_overlap \leftarrow \max(0, \min(right1, right2) - \max(left1, left2))$
25:     $y\_overlap \leftarrow \max(0, \min(bottom1, bottom2) - \max(top1, top2))$
26:     **Return** $x\_overlap \times y\_overlap$
27: **end procedure**

28: **procedure** CALCULATE EUCLIDEAN DISTANCE BETWEEN CENTROIDS OF BBOXES
29:     **Require** $centroid1, centroid2$
30:     $diff_x \leftarrow centroid2[0] - centroid1[0]$
31:     $diff_y \leftarrow centroid2[1] - centroid1[1]$
32:     $distance \leftarrow \sqrt{diff_x^2 + diff_y^2}$
33:     **Return** $distance$
34: **end procedure**

35: **procedure** CHECK THE CLOSE PROXIMITY OF A TARGET BOUNDING BOX TO OTHER BOXES IN A VIDEO FRAME
36:     **Require** $target\_bbox, bboxes\_list$
37:     $best\_distance \leftarrow \infty$
38:     $best\_match \leftarrow \text{None}$
39:     **for** $bbox$ in $bboxes\_list$ **do**
40:         $distance \leftarrow \text{euclidean\_distance}(bbox, target\_bbox)$
41:         **if** $distance < THRESHOLD$ **then**
42:             $best\_distance \leftarrow distance$
43:             $best\_match \leftarrow bbox$
44:             **Return** $True$
45:         **end if**
46:     **end for**
47:     **Return** $False$
48: **end procedure**

---

## 4. Implementation

### 4.1. Dataset Preprocessing

We implemented various image preprocessing techniques to the training dataset to improve the training accuracy and decrease training loss of the ML models. First, we changed the color contrast of the images and applied Gaussian noise. Also, we used image desaturation, making pixel colors more muted by adding more black and white colors. These transformations aim to reduce the influence of the background factor during the training process.

Before training the ML models, we normalized the dataset's pixel intensity values of cow images, distributed as a Gaussian curve centered at zero. Image normalization was calculated by subtracting the mean value of the cow image $\delta$ from the value of each pixel $C(i, j)$, then dividing the output by the cow image's standard deviation $\alpha$, as follows:

$$X(i,j) = \frac{C(i,j) - \delta}{\alpha} \tag{12}$$

where $C$ is the input cow image, $X$ is the output image, and $i$ and $j$ are the current pixel indices to be normalized.

We augmented the number of images for a few cow clusters that lack a training set to avoid the overfitting issue of the ML models. Figure 5 illustrates a sample of the implemented geometric transformations applied to these images. In particular, we implemented horizontal flipping, $-45°$ to $45°$ rotation, $1.5\times$ scaling, zoom with a range of 0.2, width and height shifts with a relative scale of 0.3, and cropping some images manually.

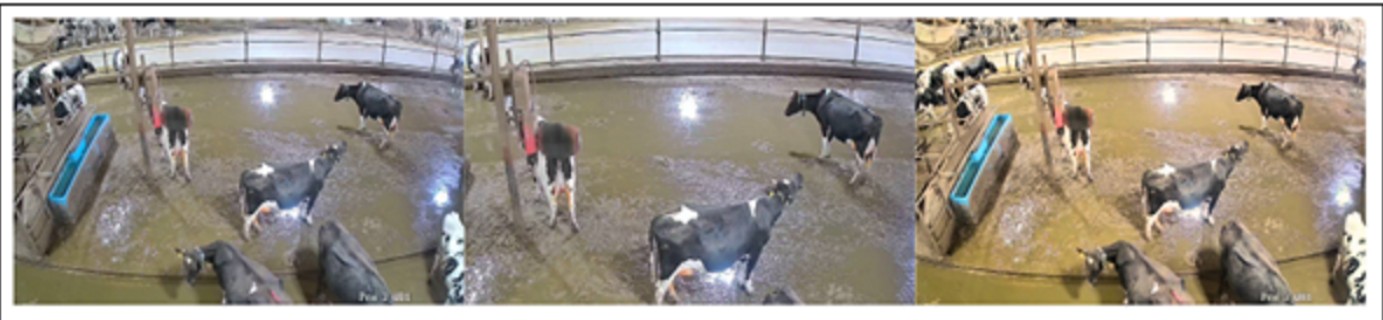

**Figure 5.** Dataset augmentation.

The Roboflow tool was used to annotate the objects of interest (i.e., cows, water banks, and brushing objects) for our detection and segmentation modules, as shown in Figure 6. Roboflow is a versatile platform for annotating the imagery dataset that utilizes the Segment Anything Model (SAM) to annotate these objects, for instance, segmentation functionality, which boosts the annotation task 10 times faster than traditional annotation methods. However, it fails to annotate some objects where the edges of two objects are mixed up. In these scenarios, we had to define and describe the spatial regions of the target objects manually.

### 4.2. ML Models

We implemented ML models using a Jupyter development environment [31]. Jupyter, which uses PyTorch [32] as a back-end engine, is an open-source neural network library written in Python. PyTorch provides comprehensive tools and pre-built functions that facilitate the construction of deep learning models. The ML model training was conducted using an Alienware server computer equipped with a 5 GHz Intel Core™ i9-16 MB CPU processor, Dell PC manufactured, Irvine, CA, USA, 2 TB SSD Hard Drive, 32 GB of RAM, and NVIDIA RTX GPU capability.

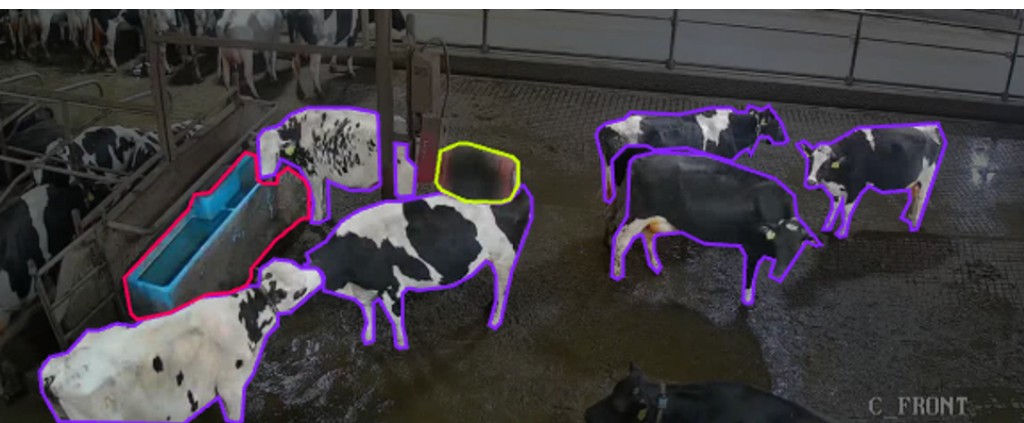

**Figure 6.** Annotating the cows, water bank and brushing objects using Roboflow.

### 4.2.1. Cow Detection and Segmentation Using YoloV8

We developed a cropping tool using the Python programming language to extract the objects of interest from the video frame images using the bounding boxes generated by YoloV8. Figure 7 shows examples of the extracted cows, water bank, and brushing objects. This process isolates the objects of interest from their background, allowing cleaner data input into the clustering phase.

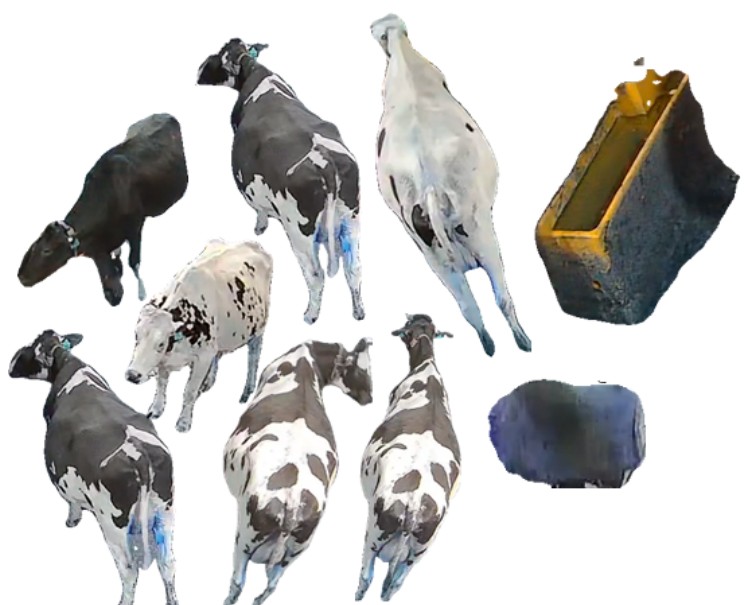

**Figure 7.** The extracted cows, water trough and brush objects.

The YOLOv8 model was trained with the annotated cow images in various poses, interactions, and lighting conditions typical of a farm setting. This diversity helps the model to recognize the cow objects reliably under different real-world conditions. We set the batch size and number of epochs to be 50 images and 100 epochs, respectively.

Figure 8 illustrates the calculated training loss of the cow detection model graphically for four different loss functions: box loss, segmentation loss, classification loss, and total loss. The accuracy increases while the Mean Squared Error loss decreases consistently over the 100 training epochs. As shown in the figure, our model converged after the 75th epoch, which means that our image dataset and the fine-tuned parameters fit the model well.

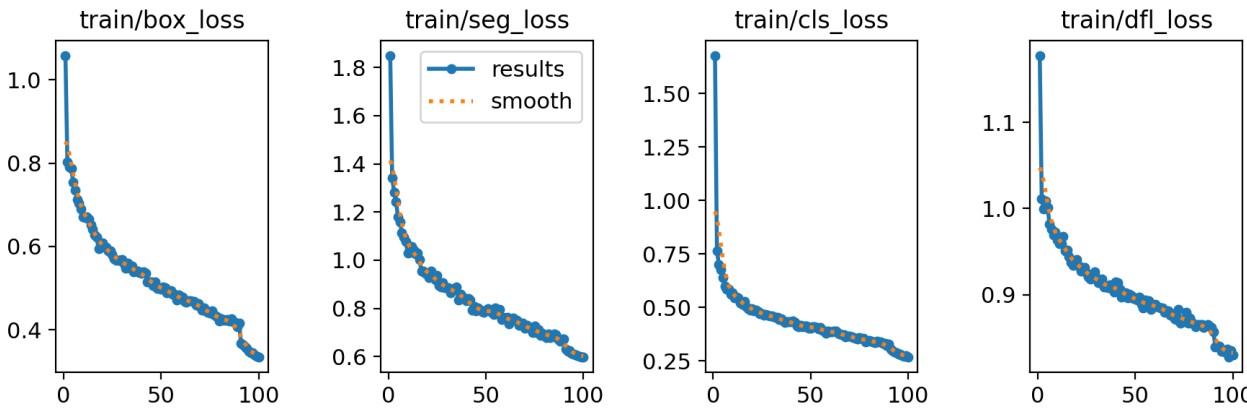

**Figure 8.** The training loss of the YoloV8 model.

Figure 9 illustrates an example inference result of the YOLOv8 model that detects the cows, water bank, and brushing objects with high accuracy. As shown in the figure, the developed YOLOv8 model performed various computer vision and ML functionalities, including object detection, segmentation, pose estimation, tracking, and classification.

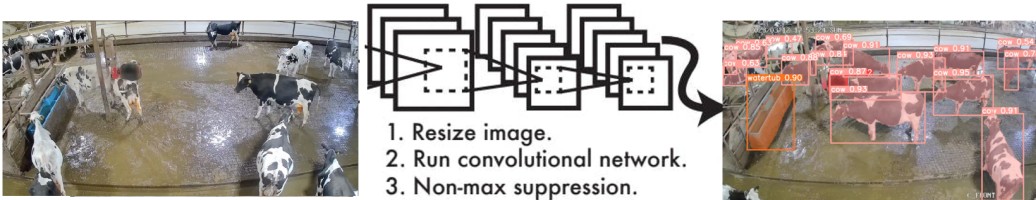

**Figure 9.** Detecting cows, water trough, and brush objects using YOLOv8.

### 4.2.2. Cow Clustering Using K-Means

The K-means clustering algorithm was implemented using scikit-learn [33], a free and open-source ML library for the Python programming language. The developed K-means model generated 300 different cow clusters, each with multiple images of the same cow from various angles and poses. Figure 10 shows an example of the cow images from the same cluster.

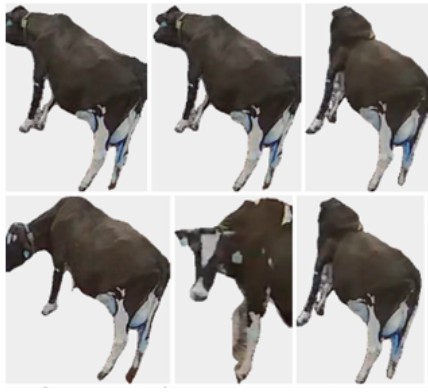

**Figure 10.** Cow objects clustering using K-means.

### 4.2.3. Cow Identification Using the CNN and SENet Model

Before training the CNN model, all cow images must be the same size. We trained the model with colored (RGB) images with resized dimensions of $200 \times 200$ pixels. We set the batch size to 100 images and the number of epochs to 15 epochs; a snapshot of the trained weights is taken every 5 epochs to monitor the progress.

The CNN model is structured with over 59 million trainable parameters. We trained the model with four fully connected convolutional layers: one input layer, a classification layer,

and SENetBlock. We adjusted several model parameters, including the learning rate, number of layers, and number of neurons per layer, to find the optimal configuration that maximizes precision and meets the confidence threshold requirements.

Figure 11 illustrates an example of the inference result of the CNN model for identifying the correct cow ID from different clusters. As shown in the figure, the CNN model classified the input cow objects to the correct cluster ID based on their coat patterns with an average accuracy of 96%.

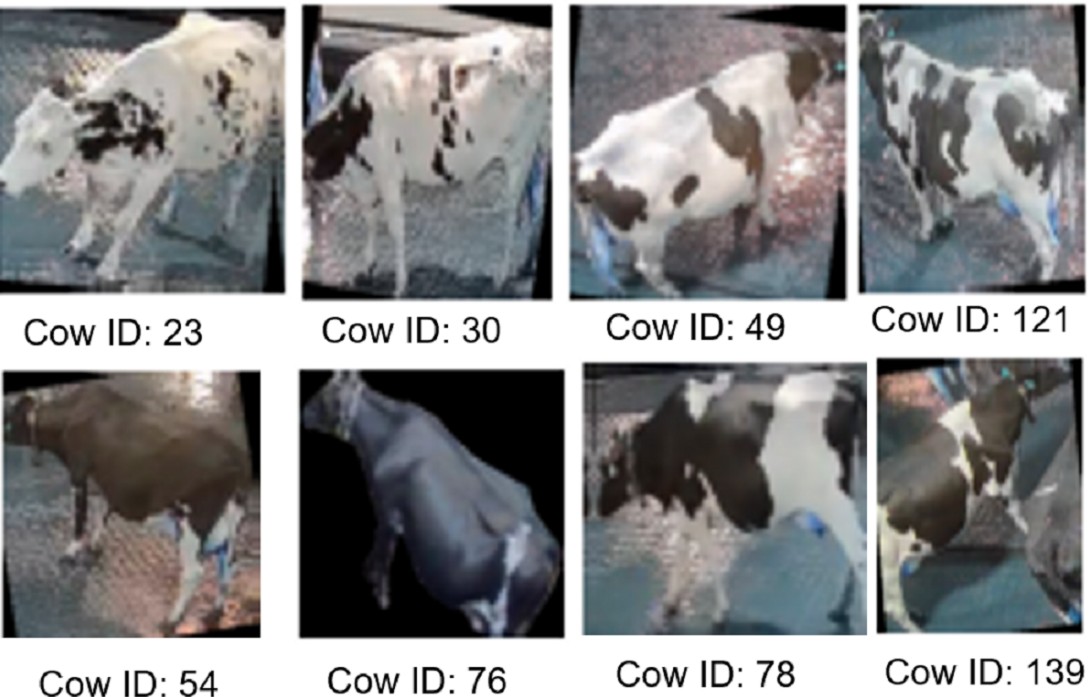

**Figure 11.** Examples of successful inference results using the CNN model.

### 4.2.4. Tracking Cow Behaviors of Interest Using DeepSORT

The DeepSORT algorithm was implemented using YOLO libraries and various Python OpenCV 4.9 libraries, including CV2, Pandas, and Shutils. Figure 12 shows successful examples of quantifying the drinking and brushing behaviors of multiple cows in the same scene. As shown in the figure, our system shows the duration of each behavior of interest in seconds, along with the identified cow IDs.

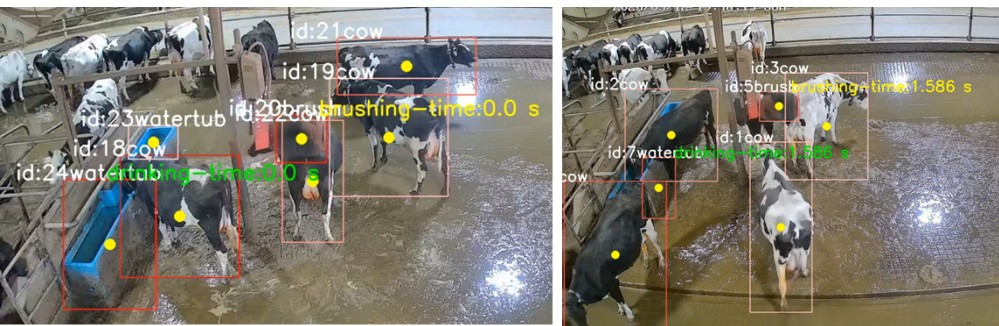

**Figure 12.** Cow behavior analysis using the overlap detection algorithm.

### 4.3. GUI User Interface

We built the web app using Python Flask Framework, ReactJS, HTML5, CSS3, JavaScript, and JSON. To run the web application on top of the ML models, we had to wrap both models, implemented on PyTorch, as a REST API using the Flask web framework. In other

words, the communication between PyTorch and Flask is coordinated through that REST API. When the user captures an image using the camera, Flask uses the `POST method to send the image from the user browser to PyTorch via an HTTP header.`

The GUI is designed to be intuitive, allowing users to easily upload videos, view analysis results, and receive notifications about the behaviors of interest related to heat stress. Figure 13a shows the videos page that allows users to upload videos and preview them to confirm correctness.

Once a video has been uploaded, the user can kick-start the cow detection pipeline by hitting the *Start Inference* button. Figure 13b shows the cows page that displays all identified cows along with their assigned identification number and sample photo. Once the inference process has been completed, the user is renavigated to the dashboard page to view the behavioral analysis completed on the uploaded video (see Figure 13c). Users can also view the duration of each behavior of interest and preview the inference video, which shows the identified cows and the duration spent.

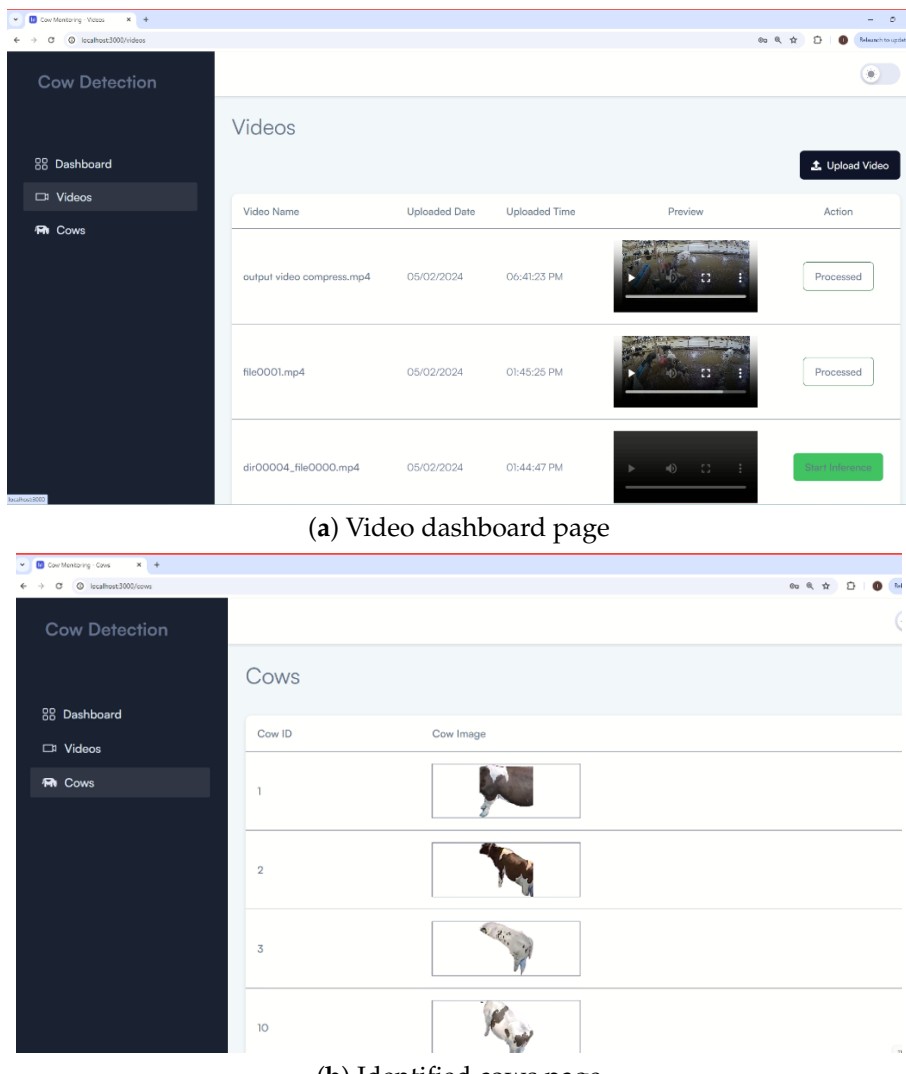

(**a**) Video dashboard page

(**b**) Identified cows page

**Figure 13.** *Cont.*

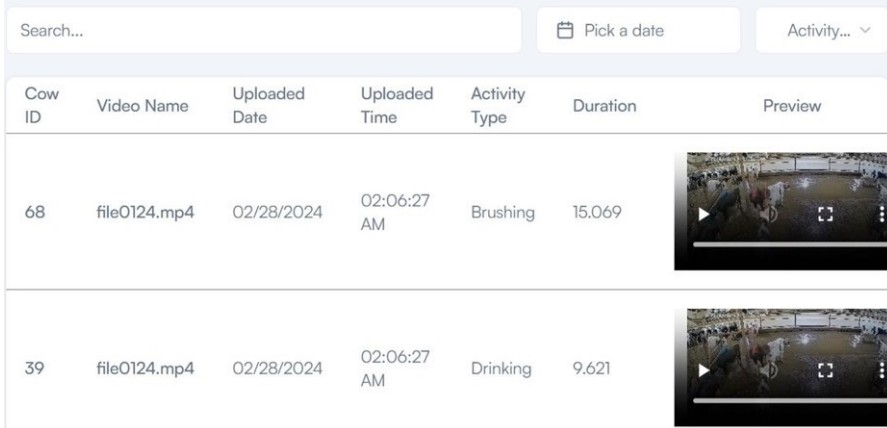

(**c**) Recognized cow behaviors page

**Figure 13.** Screenshots of the web-based GUI user interface. (**a**) The video dashboard page that allows users to upload the videos for inference. (**b**) The cow's page that shows the identified cow IDs that appeared in the videos. (**c**) The recognized cow behaviors page showing the drinking and brushing activities along with their timestamps and durations.

## 5. Evaluation

We experimentally evaluated our prototype implementation regarding classification accuracy and performance. For classification accuracy, we observed that our system delivers good results in natural conditions even when the images are captured from different distances from the camera, orientations, and illumination conditions. Figure 14 shows an example of successful inference of cow identifiers and their behaviors of interest, along with the duration spent in each activity.

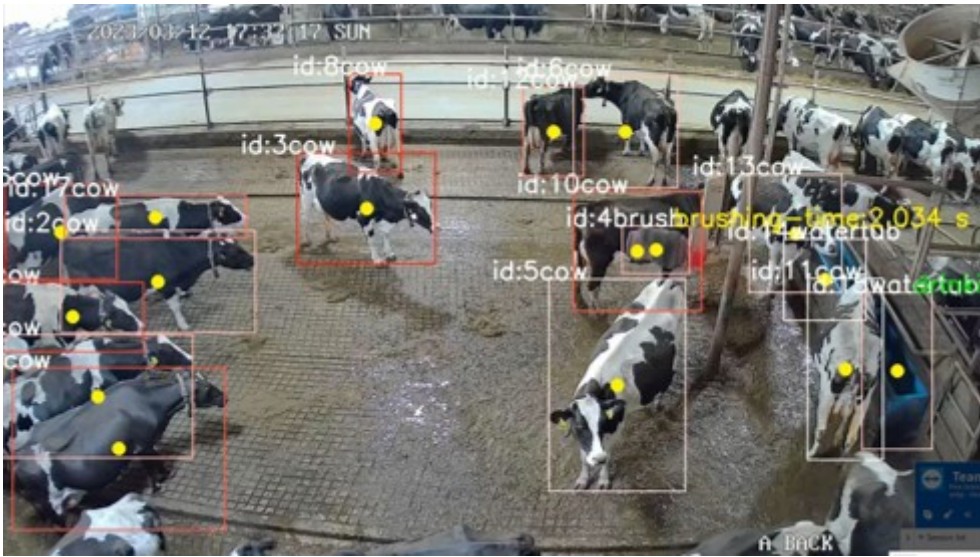

**Figure 14.** An Example of successful inference of cow identifiers and their behaviors of interest.

The precision vs. recall curve, shown in Figure 15, summarizes the trade-off between the true positive rate and the positive predictive value for our YoloV8 model using different probability thresholds. In other words, it indicates the model's ability to accurately identify the cow objects while maintaining a balance between false positives and false negatives. The curve demonstrates that the model achieves high precision and recall across a wide range of thresholds. Also, it attests to its effectiveness in detecting cows regardless of the sensitivity level, which proves that our system can be reliably deployed in real-world scenarios.

Precision represents the positive predictive value of our model, while recall is a measure of how many true positives are identified correctly. As shown in the figure, the

precision vs. recall curve tilts towards 1.0, which means that our YoloV8 model achieves high accuracy while minimizing the number of false negatives.

The precision ratio describes the performance of our model at predicting the positive class. It is calculated by dividing the number of true positives (*TPs*) by the sum of *TPs* and false positives (*FPs*), as follows:

$$Precision = \frac{TPs}{TPs + FPs} \tag{13}$$

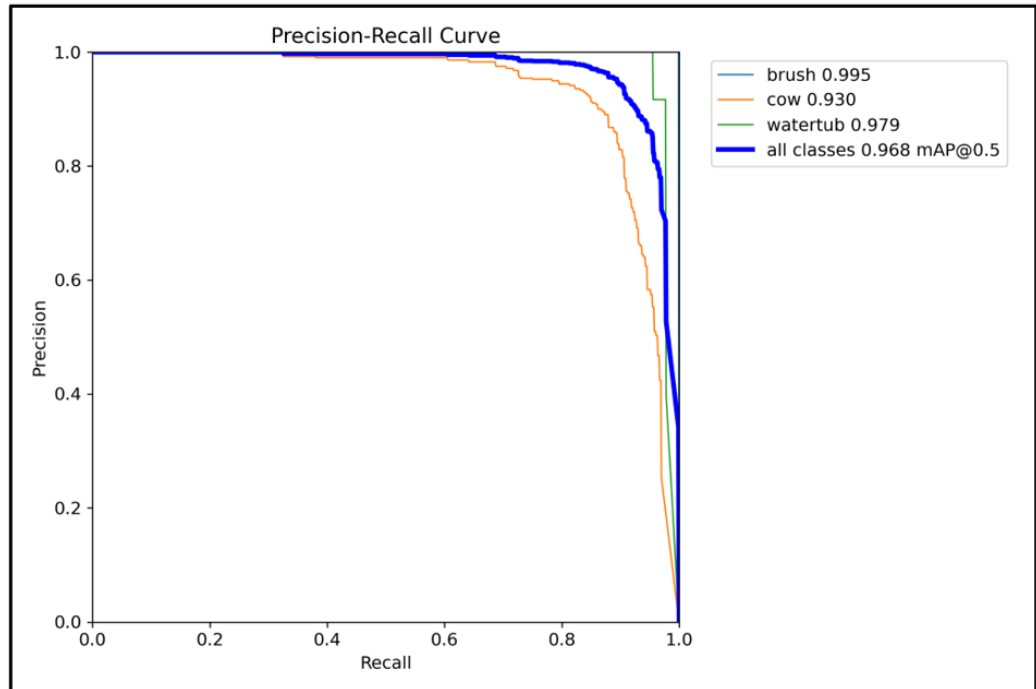

**Figure 15.** The precision–recall curve of the YoloV8 model.

The recall ratio is calculated as the ratio of the number of true positives divided by the sum of *TPs* and the false negatives (*FNs*), as follows:

$$Recall = \frac{TPs}{TPs + FNs} \tag{14}$$

The overall classification accuracy of our model is calculated as the ratio of correctly predicted observation (i.e., the sum of TPs and true negatives (*TNs*)) to the total observations (i.e., the sum of *TPs*, FPs, FNs, and *TNs*) using this equation:

$$Accuracy = \frac{TPs + TNs}{TPs + FPs + FNs + TNs} \tag{15}$$

The YoloV8 model achieved an overall average classification accuracy of 93%, 97.9%, and 99.5% for the cow, water tub, and brush tool objects, respectively. The CNN model for cow identification achieved an overall average classification, recall, and F1-score of 96%, 97%, and 97%, respectively.

## 6. Conclusions and Future Work

As intensive dairies grow, the need for automatic cattle monitoring becomes more pressing. Manual observation can be practical on a small scale but quickly becomes infeasible when dairies host hundreds or thousands of cows. Further, there is an increasing need to use modern technologies, including computer vision and AI, to track behavioral changes to alert the farmer of the herd's health status. This paper presented the design and

implementation of an ML-powered approach for automatically characterizing behavioral phenotypes for dairy cows relevant to thermotolerance.

We collected a dataset consisting of 3584 videos of 24 h of continuous recording of hundreds of cows captured from the T&K Dairy in Snyder, Texas. The developed system used computer vision and ML models to monitor two cow behaviors of interest: the drinking and brush use of dairy cows in a robotic milking system. In particular, we utilized the YoloV8 model to detect and segment cow, water tub, and brushing tool objects. The K-means algorithm is used to group the cows into clusters, which is used as input to a CNN model to identify the cows in the videos. We used the DeepSORT model to track the cow activities in the barn. We finally quantified the behaviors of interest using the developed overlap detection algorithm. A user-friendly interface was created on top of the ML models, allowing ranchers to interact with the system conveniently.

We tested our system with a dataset of various cow videos, where crowded backgrounds, low contrast, and images of diverse illumination conditions were considered. Our system achieved high precision in object detection and behavior recognition, which was corroborated by the system's ability to accurately track and analyze the cow behaviors of interest within a dynamic farm environment. Most notably, the YoloV8 and CNN models achieved accuracies of 93% and 96% in detecting the objects of interest and identifying the cow IDs, respectively.

In ongoing work, we are looking into opportunities for generalizing our approach to detect a broader range of changes in behaviors or health indicators in various farm conditions [6], such as increased mounting or standing behavior that can indicate that a cow is going into estrus. In contrast, changes in walking and lying behavior can indicate lameness before it is evident enough to be noticed by manual inspection [21]. Another avenue of further improvement is incorporating IoT sensors into the barn that could automate data collection and action initiation, such as adjusting environmental conditions in response to detected behaviors, thereby enhancing the system's responsiveness. We expect the developed system to inform the genetic selection decisions and impact dairy cow welfare and water use efficiency.

**Author Contributions:** Conceptualization, A.A.A.; methodology, O.I.; software, O.I.; validation, G.M.; formal analysis, A.A.A.; investigation, O.I.; resources, C.D.; data curation, G.M.; writing—original draft preparation, A.A.A., G.M. and O.I.; writing—review and editing, A.A.A. and C.D.; visualization, O.I. and G.M.; supervision, A.A.A.; project administration, A.A.A. and C.D.; funding acquisition, A.A.A. and C.D. All authors have read and agreed to the published version of the manuscript.

**Funding:** This research work is supported in part by the National Science Foundation (NSF) under grants #2011330 and 2200377. Any opinions, findings, and conclusions expressed in this paper are those of the authors and do not necessarily reflect NSF's views.

**Institutional Review Board Statement:** Not applicable.

**Informed Consent Statement:** Not applicable.

**Data Availability Statement:** The data and source code that support the findings of this study are available online. Source Code: https://zenodo.org/doi/10.5281/zenodo.12773943 accessed on 30 July 2024. Dataset: https://zenodo.org/doi/10.5281/zenodo.12627650 accessed on 30 July 2024.

**Conflicts of Interest:** The authors declare no conflicts of interest.

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
