# Peer review of "On Developing a Machine Learning-Based Approach for the Automatic Characterization of Behavioral Phenotypes for Dairy Cows Relevant to Thermotolerance"

_agriengineering, doi:10.3390/agriengineering6030155_

Round 1

Reviewer 1 Report

Comments and Suggestions for Authors

The article presents an innovative approach utilizing machine learning (ML) and computer vision techniques to quantify behavioral indicators of thermotolerance in dairy cows. The focus is on drinking frequency and brush use behaviors, which are challenging to measure through traditional methods. The methodology includes using CNN and YOLOv8 for object detection and DeepSORT for tracking behaviors. The system aims to assist ranchers in mitigating heat stress impacts and enhancing dairy cow welfare.

1) Include more recent and relevant studies to provide a comprehensive context and justification for your chosen methods.

2) Clearly articulate how your approach advances beyond existing methods. A comparison table summarizing differences and improvements would be beneficial.

3) Discuss the variability in cow behavior across different environments and suggest ways to adapt the model to diverse settings. Future work should include testing in various farm conditions.

4) Provide more comprehensive details on model configurations, training processes, and parameter tuning to enable replication and validation of the results.

5) Include user feedback or usability testing results. Detailed case studies or examples of how ranchers have benefited from using the system would strengthen the application aspect.

Comments on the Quality of English Language

1) Ensure consistent use of technical terms throughout the paper. For instance, "thermotolerance" and "heat stress tolerance" should be used consistently to avoid confusion.

2) Some sentences are overly complex and could benefit from being broken down into simpler sentences.

3) Correct minor grammatical errors, such as missing articles or incorrect verb tenses.

Author Response

Concern # 1: Include more recent and relevant studies to provide a comprehensive context and justification for your chosen methods.

Author response:  Thanks for the comment. We updated the related work section to address this point. Also, in the revised manuscript, we clearly articulated the motivations and advantages of the selected methods.

Concern # 2: Clearly articulate how your approach advances beyond existing methods. A comparison table summarizing differences and improvements would be beneficial.

Author response: Thanks for the comment. We updated the related work section to highlight the advancement of our approach in comparison to the existing work. For instance, “Despite the previous success of cattle identification using computer vision to identify coat patterns, the existing work has notable limitations. The used imagery dataset has a small field of view, often set up where cattle walk through narrow passages with limited ability to turn. Additionally, the lighting is constant, and there may be only one or a few cattle in the frame at a time, all of which simplify the task of identifying cattle by computer vision but limit the potential applications in a busy barn. In contrast, our approach is designed to identify cattle at a distance and in an open space within a broad view frame.”

Concern # 3: Discuss the variability in cow behavior across different environments and suggest ways to adapt the model to diverse settings. Future work should include testing in various farm conditions.

Author response: Thanks for the comment. The revised manuscript discussed the variability in cow behavior within complex farm environments. Also, in the future work section, we added a new paragraph for generalizing our approach to detect a broader range of changes in behaviors or health indicators in various farm conditions, such as increased mounting or standing behavior that can indicate that a cow is going into estrus.

Concern # 4: Provide more comprehensive details on model configurations, training processes, and parameter tuning to enable replication and validation of the results.

Author response: Thanks for the comment. We added details on the ML model configurations to allow replication of the results. Also, the source code for all models and GUI is available on request for any researcher and practitioner working in the field.

Concern # 5: Include user feedback or usability testing results. Detailed case studies or examples of how ranchers have benefited from using the system would strengthen the application aspect.

Author response: Thanks for the comment. Although the developed GUI is still a proof-of-concept demo subject to testing by ranchers, we plan to incorporate user usability testing in the project's second phase.

Concern # 6: Ensure consistent use of technical terms throughout the paper. For instance, "thermotolerance" and "heat stress tolerance" should be used consistently to avoid confusion.

Author response: Thanks for the comment. We revised the manuscript to address this point. We used the term “thermotolerance” consistently throughout the paper.

Concern # 7: Some sentences are overly complex and could benefit from being broken down into simpler sentences.

Author response: Thanks for the comment. We updated the manuscript to reduce the usage of complex sentences. We explained every term used in the paper to enable general readers to understand the technical content of this work.

Concern # 8: Correct minor grammatical errors, such as missing articles or incorrect verb tenses.

Author response: Thanks for the comment. We updated the manuscript by fixing the grammatical and editing errors and drastically enhancing the general presentation of the paper, including graphics, discussion, and exposition.

Reviewer 2 Report

Comments and Suggestions for Authors

Please explain in more detail how recall can be improved, that is, input for false negative items of certain indicators can be reduced.

Author Response

Concern # 1: Please explain in more detail how recall can be improved, that is, input for false negative items of certain indicators can be reduced.

Author response: Thanks for the comment. The Recall metric measures how many true positives are identified correctly. We added new sentences to describe how the recall value can be improved by achieving high accuracy while minimizing the number of false negatives.

Reviewer 3 Report

Comments and Suggestions for Authors

This is a paper that uses a deep learning network to detect cow behavior. Instead of being a research paper, the reviewer believes it is more like a research process report. Without discussing the originality of the paper, just in terms of article structure and content, the reviewer believes it needs to be revised, as follows:

1. The abstract section only narrates the research background and significance, without specifying the test methods, test data, and test conclusions.

2. The introduction and related work sections contain too much discussion and should be merged. This paper is a research paper, not a review.

3. There are too many unnecessary figures in the content. The program code does not need to be listed in its entirety; one schematic diagram explaining the author’s improvements to the Yolo network is sufficient. Additionally, many on-site images are not necessary to include in the paper. The program structure diagram is not ideal. The GUI interface does not need to be included in the paper.

4. The paper only conducts simple test and lacks discussion. Overall, it is more like an instruction manual for the Yolo algorithm improvement process.

Comments on the Quality of English Language

None.

Author Response

Concern # 1: The abstract section only narrates the research background and significance, without specifying the test methods, test data, and test conclusions.

Author response: Thanks for the comment. We updated the abstract to address this point. We added the used methods and performance evaluation results.

Concern # 2: The introduction and related work sections contain too much discussion and should be merged. This paper is a research paper, not a review.

Author response: Thanks for the comment. We summarized the introduction and related work sections. Also, we updated the related work section to highlight the advancement of our approach compared to the existing work.

Concern # 3: There are too many unnecessary figures in the content. The program code does not need to be listed in its entirety; one schematic diagram explaining the author’s improvements to the Yolo network is sufficient. Additionally, many on-site images are not necessary to include in the paper. The program structure diagram is not ideal. The GUI interface does not need to be included in the paper.

Author response: Thanks for the comment. We included all these figures to illustrate our pipeline approach for processing the video dataset, where data flows from one layer to the next in detail. These figures show the different phases of cow Detection, clustering, identification, and tracking behaviors of interest. This allows other researchers to replicate our approach. We updated the manuscript to remove some figures that were clearly described in the text.

Concern # 4: The paper only conducts simple test and lacks discussion. Overall, it is more like an instruction manual for the Yolo algorithm improvement process.

Author response: Thanks for the comment. To the best of our knowledge, our approach represents the first step to building a system that automatically identifies dairy cattle based on biometric features and monitors their behavior of interest based on interactions with other objects in the barn (i.e., water troughs and brush stations). We updated the introduction section to mention the contributions of this paper. “The contributions of this paper are threefold. First, we propose an ML-based approach capable of capturing, processing, and visualizing large video datasets to characterize behavioral phenotypes for dairy cows relevant to thermotolerance automatically. Second, a novel object-tracking module is proposed for detecting the behavior of moving cows in surveillance videos in real-time. It is designed to be generic, making it applicable to different fields requiring real-time processing using CCTV footage videos in agriculture. Third, we developed a user-friendly interface on top of a pipeline of ML models and computer vision algorithms (i.e., K-means, YoloV8, CNN, and DeepSORT) to allow ranchers to interact with the developed system conveniently using a web GUI interface.”

Round 2

Reviewer 3 Report

Comments and Suggestions for Authors

None.

Author Response

Thank you for your comments. We addressed all these points in the revised version.